# Spectrum of Atazanavir-Selected Protease Inhibitor-Resistance Mutations

**DOI:** 10.3390/pathogens11050546

**Published:** 2022-05-05

**Authors:** Soo-Yon Rhee, Michael Boehm, Olga Tarasova, Giulia Di Teodoro, Ana B. Abecasis, Anders Sönnerborg, Alexander J. Bailey, Dmitry Kireev, Maurizio Zazzi, Robert W. Shafer

**Affiliations:** 1Department of Medicine, Stanford University, Stanford, CA 94305, USA; alexjamesbailey@gmail.com (A.J.B.); rshafer@stanford.edu (R.W.S.); 2Institute of Virology, Faculty of Medicine and University Hospital of Cologne, University of Cologne, 50935 Cologne, Germany; michael.boehm@uk-koeln.de; 3Department of Bioinformatics, Institute of Biomedical Chemistry, 119121 Moscow, Russia; olga.a.tarasova@gmail.com; 4Department of Computer, Control and Management Engineering Antonio Ruberti, Sapienza University of Rome, 00185 Rome, Italy; giulia.diteodoro@uniroma1.it; 5Global Health and Tropical Medicine, Instituto de Higiene e Medicina Tropical, Universidade Nova de Lisboa, 1349-008 Lisboa, Portugal; ana.abecasis@ihmt.unl.pt; 6Division of Infectious Diseases, Department of Medicine Huddinge, Karolinska Institute, Huddinge, 14186 Stockholm, Sweden; anders.sonnerborg@ki.se; 7Central Research Institute of Epidemiology, 111123 Moscow, Russia; dmitkireev@yandex.ru; 8Department of Medical Biotechnologies, University of Siena, 53100 Siena, Italy; maurizio.zazzi@unisi.it

**Keywords:** HIV-1, antiviral therapy, drug resistance, protease inhibitor, protease, mutation, atazanavir

## Abstract

Ritonavir-boosted atazanavir is an option for second-line therapy in low- and middle-income countries (LMICs). We analyzed publicly available HIV-1 protease sequences from previously PI-naïve patients with virological failure (VF) following treatment with atazanavir. Overall, 1497 patient sequences were identified, including 740 reported in 27 published studies and 757 from datasets assembled for this analysis. A total of 63% of patients received boosted atazanavir. A total of 38% had non-subtype B viruses. A total of 264 (18%) sequences had a PI drug-resistance mutation (DRM) defined as having a Stanford HIV Drug Resistance Database mutation penalty score. Among sequences with a DRM, nine major DRMs had a prevalence >5%: I50L (34%), M46I (33%), V82A (22%), L90M (19%), I54V (16%), N88S (10%), M46L (8%), V32I (6%), and I84V (6%). Common accessory DRMs were L33F (21%), Q58E (16%), K20T (14%), G73S (12%), L10F (10%), F53L (10%), K43T (9%), and L24I (6%). A novel nonpolymorphic mutation, L89T occurred in 8.4% of non-subtype B, but in only 0.4% of subtype B sequences. The 264 sequences included 3 (1.1%) interpreted as causing high-level, 14 (5.3%) as causing intermediate, and 27 (10.2%) as causing low-level darunavir resistance. Atazanavir selects for nine major and eight accessory DRMs, and one novel nonpolymorphic mutation occurring primarily in non-B sequences. Atazanavir-selected mutations confer low-levels of darunavir cross resistance. Clinical studies, however, are required to determine the optimal boosted PI to use for second-line and potentially later line therapy in LMICs.

## 1. Introduction

Ritonavir-boosted atazanavir has become increasingly important as an option for second-line therapy in low- and middle-income countries (LMICs) [1]. Although it appears to have comparable efficacy to ritonavir-boosted lopinavir (lopinavir/r) [2,3], there are few data on the mutations arising in patients receiving boosted or unboosted atazanavir compared with the extensive data available for lopinavir/r [4,5,6,7,8,9,10,11,12]. Characterizing the spectrum of mutations arising in patients receiving atazanavir, whether boosted or unboosted, provides an insight into the genetic barrier to atazanavir resistance and into the use of boosted darunavir (darunavir/r) for third line therapy in LMICs.

Therefore, in this paper, we analyze publicly available protease sequences from previously protease inhibitor (PI)-naïve patients with virological failure (VF) on a boosted or unboosted atazanavir-containing regimen. We compare the spectrum of protease mutations observed in patients with subtype B as opposed to non-B viruses, in patients receiving boosted as opposed to unboosted atazanavir, and in patients with early PI resistance (e.g., harboring few PI-associated drug-resistance mutations (DRMs)) with advanced PI resistance (e.g., harboring four or more PI-associated DRMs). We also examine the predicted susceptibility of the different patterns of atazanavir-selected mutations to lopinavir/r and darunavir/r.

## 2. Results

### 2.1. Studies

Overall, 1763 protease sequences from 1497 patients reported in 30 studies who received either boosted or unboosted atazanavir as their first PI were available for the analysis (Table 1). These sequences included 773 sequences from 740 patients in 27 studies from Stanford HIV Drug Resistance Database (HIVDB) [13], and previously unpublished sequences, including (i) 741 sequences from 562 patients from the EuResist Integrated Database (EIDB) [14]; (ii) 206 sequences from 152 patients from the Stanford University Hospital (SUH); and (iii) 43 sequences from 43 patients from the RHIVDB [15], a freely accessible database of HIV-1 sequences and clinical data of infected patients. Of the 184 patients with more than 1 sequence, 17 had sequences that differed from one another by one or more DRMs. For these patients, we selected the sequence containing the largest number of PI-associated DRMs. The complete set of 1497 one-per-person HIV-1 group M sequences from persons receiving atazanavir was provided in Appendix A.

The 30 studies were published between 2004 and 2021. The median number of patients per study was 12 (IQR: 1–39). The distribution of studies and patients by region included Africa (10 studies; 119 patients), North America (5 studies; 383 patients), Europe (4 studies; 565 patients), Latin America (4 studies; 199 patients), Eastern Europe (3 studies; 55 patients), and Asia (2 studies; 38 patients). Two studies included 138 patients from 1 or more regions.

The median sample year was 2011 (IQR: 2007–2015). Approximately 99% of sequences were obtained from plasma and 1% from peripheral blood mononuclear cells (PBMCs). Next-generation sequencing (NGS) was performed in 1 of the 30 studies. The most common subtypes were B (61.9%), C (13.6%), A (6.7%), G (6.1%), 02_AG (4.9%), F (3.1%) and D (1.1%). Of 1497 patients, 62.7% (n = 939) received boosted atazanavir and 37.3% (n = 558) received unboosted atazanavir. A higher proportion of patients with subtype B (70.4% of 927) compared with non-subtype B (50.2% of 570) viruses received boosted (*p* < 0.001). Table 2 summarizes the numbers of patients according to the administration of atazanavir (boosted vs. unboosted), subtype (B vs. non-subtype B), previous antiretroviral therapy (ART) (naïve vs. experienced), and year of ART initiation.

### 2.2. Mutation Prevalence

Of the 1497 patients, 264 (17.6%) had 1 or more PI-associated DRMs. Of the 57 HIVDB PI-associated DRMs, 48 occurred in ≥1 patient, 38 in ≥2 patients, and 24 in ≥5 patients. The most commonly occurring major DRMs were I50L (34.1%), M46I (32.6%), V82A (22.3%), L90M (19.3%), I54V (16.3%), N88S (10.2%), M46L (7.6%), V32I (6.4%), and I84V (6.1%) (Table 3). The most common accessory DRMs were L33F (20.8%), Q58E (15.9%), K20T (14.4%), G73S (11.7%), L10F (9.8%), F53L (9.8%), K43T (8.7%), and L24I (6.1%).

Of the 264 sequences with 1 or more PI-associated DRMs, the proportions of the sequences containing 1 DRM, 2–3 DRMs and ≥4 DRMs were 33.7%, 31.4% and 34.9%, respectively. The distribution of DRMs differed according to the total number of DRMs per sequence (Figure 1). Among sequences with a single DRM, the most common major DRMs were I50L, M46I/L, L90M, and N88S, while the most common accessory DRMs were Q58E, K20T, G73S, and L33F. In contrast, among sequences with ≥4 DRMs, the most common major DRMs were M46I/L, V82A, L90M, I54V, I50L, and N88S, while the most common accessory DRMs were unchanged. The major DRMs V32I and I84V occurred in approximately 5% to 6% of sequences regardless of the total number of DRMs.

An additional 197 mutations, previously classified as nonpolymorphic treatment selected mutations (NP-TSMs), occurred in 149 sequences, including in 109 of the 264 sequences containing a PI-associated DRM and 40 of the 1215 sequences without a PI-associated DRM. There were 33 different NP-TSMs of which the most common were L89T (34.9% of 149 sequences), K55R (15.4%), I85V (11.4%), A71I (9.4%), and E34Q (7.4%) (Appendix A). These mutations were not classified as DRMs because they do not receive an HIVDB mutation penalty score.

### 2.3. Unboosted versus Boosted Atazanavir

PI-associated DRMs occurred in 21.0% of 558 patients receiving unboosted atazanavir and 15.7% of 939 patients receiving boosted atazanavir (*p* = 0.01) (Table 2). However, among patients with a DRM, the median number of DRMs was not significantly greater in those receiving unboosted atazanavir (3 DRMs; IQR: 1–4) compared with boosted atazanavir (2 DRMs; IQR: 1–4; *p* = 0.1). Of the 48 reported DRMs, I50L was the only DRM that occurred more commonly in patients receiving unboosted as compared with boosted atazanavir (10.4% vs. 3.4%; adjusted *p* < 0.001).

Sequences from patients receiving unboosted atazanavir were also slightly more likely to have one or more NP-TSMs compared with sequences from patients receiving boosted atazanavir (12.2% of 558 vs. 8.6% of 939; *p* = 0.03). Each of the 33 reported NP-TSMs occurred in similar proportions in patients receiving unboosted and boosted atazanavir.

### 2.4. Subtypes

The proportion of sequences containing one or more PI-associated DRMs was similar in subtype B (20.0% of 570) versus non-subtype B (16.2% of 927; *p* = 0.07) sequences (Table 2). Of the 48 reported PI-associated DRMs, G73S was significantly more common in subtype B (3.1% of 927) than non-subtype B (0.4% of 570; adjusted *p* = 0.005) sequences. Of the 33 reported NP-TSMs, only L/M89T was significantly more common in non-subtype B than in subtype B sequences (8.4% of 570 vs. 0.4% of 927; adjusted *p* < 0.001). In subtypes A, C, G, CRF01_AE, and CRF02_AG, the consensus amino acid at position 89 is methionine (M) [41] and 89T requires just a single transition in these subtypes (ATG => ACG). In contrast, changing to 89T requires a one transition plus one transversion change in subtype B (CTN or TTR => ACN).

### 2.5. ART Experience

Among the 1497 patients receiving atazanavir, 907 (60.6%) were previously ART-naïve and 590 (39.4%) were ART-experienced (Table 2). The proportion of sequences containing one or more PI-associated DRMs was 21.7% in previously ART-experienced patients and 15.0% in previously ART-naïve patients (*p* = 0.001). Among those with one or more PI-associated DRMs, the number of DRMs was not significantly different in previously ART-experienced patients (median 2 DRMs; IQR: 1–4 DRMs) compared with previously ART-naïve patients (median 3 DRMs; IQR: 1–4 DRMs; *p* = 0.3).

Among the 907 previously ART-naïve patients, atazanavir was administered with 2 nucleoside RT inhibitors (NRTIs) in 840 (92.6%) patients. Among the remaining 67 patients, the co-administered antiretroviral drugs (ARVs) were not provided for 44 (4.9%), while 23 (2.5%) received a variety of other ARVs.

Among the 590 previously ART-experienced patients, atazanavir was administered with 2 NRTIs in 345 (58.5%) patients. Among the remaining 245 patients, the co-administered ARVs were not provided for 163 (27.6%), while 82 (13.9%) received a variety of other ARVs. Only four patients received atazanavir plus one additional ARV.

The year of ART-initiation was available for 1127 (75.3%) of all patients. The patients could be pooled into four time periods containing approximately equal numbers spanning the years between 1993 and 2018 (Table 2). The proportion of patients with one or more PI-associated DRMs decreased over time (binomial coefficient = −0.26; 95% CI: −0.45 to −0.07; *p* = 0.007), but the number of DRMs in patients with one or more DRMs did not change.

Using just those patients for whom the year of ART initiation was available, a multivariate logistic regression analysis was performed to assess the association between four factors and the development of a PI-associated DRM. The four factors included the year of ART initiation, subtype (B vs. non-subtype B), the use boosted vs. unboosted atazanavir, and previous ART (naïve vs. experienced). The analysis found that a later year of ART initiation (OR: 0.62; 95%CI: 0.49–0.79; *p* = 0.0001) and the administration of boosted atazanavir (OR: 0.57; 95%CI: 0.35–0.93; *p* = 0.02) were associated with a decreased risk of developing a PI-associated DRM.

### 2.6. Bayesian Network Analysis of Correlated Mutations

We used the 1437 (96%) sequences containing 0 to 4 PI-associated DRMs (i.e., sequences with ≥5 PI-associated DRMs were excluded) to calculate Jaccard similarity coefficients and their standard Z scores for all pairs of DRMs and NP-TSMs. Eleven pairs of mutations comprising six major DRMs (M46I, I50L, I54V, V82A, N88S and L90M), three accessory DRMs (K20T, L33F and G73S), and the NP-TSM L89T participated in one or more significant pairwise correlations (*p* < 0.01). We then performed a Bayesian network analysis to determine the conditional dependency between the mutations in each of the pairwise correlations (Figure 2).

### 2.7. Estimated Cross Resistance to LPV/r and DRV/r

Among the 264 sequences with 1 or more PI-associated DRMs, there were 182 distinct DRM patterns, including 124 patterns (164 sequences; 62.1% of 264) interpreted by HIVDB as causing high-level atazanavir resistance, 19 patterns (20 sequences; 7.6% of 264) as causing intermediate atazanavir resistance, and 29 patterns (51 sequences; 19.3% of 264) as causing low- or potential low-level atazanavir resistance. The remaining 10 DRM patterns (n = 29 sequences patterns; 11.0% of 264) consisting primarily of singe accessory DRMs (e.g., K20T, Q58E) were not interpreted as causing reduced atazanavir susceptibility.

A total of 56 distinct DRM patterns (58 sequences; 22.0% of 264) were interpreted as causing high-level lopinavir resistance, 40 patterns (43 sequences; 16.3% of 264) as causing intermediate lopinavir resistance, and 44 patterns (62 sequences; 23.5% of 264) as causing low- or potential low-level lopinavir resistance. A total of 3 distinct DRM patterns (3 sequences; 1.1% of 264) were interpreted as causing high-level darunavir resistance, 14 patterns (14 sequences; 5.3% of 264) as causing intermediate darunavir resistance, and 32 patterns (34 sequences; 12.9% of 264) as causing low- or potential low-level darunavir resistance.

### 2.8. Virological Failure with Resistance

Five of the thirty studies included participants from three clinical trials and from two clinical cohorts for which genotypic resistance testing was routinely available (Table 1). Together, these five studies included 1037 (69.3%) of all 1497 patients from whom sequences were available. Of these 1037 patients, 63.0% and 37.0% received boosted and unboosted atazanavir, respectively. In these studies, the proportion of sequences containing one or more PI-associated DRMs ranged from 2.9% to 14.5% and the overall proportion of sequences containing one or more PI-associated DRMs in patients receiving boosted and unboosted atazanavir were 7.2% and 13.5%, respectively.

### 2.9. Studies Not Included in the Analysis

We identified 32 additional studies reporting sequences from 1089 previously PI-naïve patients receiving boosted or unboosted atazanavir-containing regimens (Appendix A). Approximately 10% of the sequences in these studies were reported to contain one or more PI-associated DRMs. However, as the sequences were not available and as different mutations were reported in different studies, we did not include the data from these studies in our analysis.

## 3. Discussion

The spectrum of atazanavir-selected mutations has been largely influenced by data published in the earliest in vitro passage experiments and clinical trials. During in vitro passage experiments with three subtype B clones, the most commonly emerging DRMs were V32I, M46I, I50L, I84V, and N88S [42]. The initial reports of the in vivo selection of PI-associated DRMs, based on the use of unboosted atazanavir in ART-naïve patients, demonstrated that I50L and G73S were the most commonly occurring mutations in patients with VF [24,43]. A few cases of VF and emergent PI-associated DRMs have been reported in the clinical trials of ART-naïve patients receiving boosted atazanavir [16,44], consistent with the hypothesis that PI-resistance mutations develop only in viruses exposed to a narrow window of suboptimal drug concentrations that both exert selective pressure on the virus and allow virus replication [45]. Nonetheless, PI resistance in previously PI-naïve patients receiving lopinavir/r for second-line therapy has increasingly been reported, usually beginning after 12–18 months of therapy [46]. In addition, phenotypic studies have shown that DRMs selected by other PIs confer atazanavir cross resistance particularly when they occur in combination [47,48].

In the years since atazanavir has been introduced, there has been a gradual accumulation of data on the spectrum of mutations emerging in previously PI-naïve patients with VF on an ART-regimen containing boosted and less commonly unboosted atazanavir. In contrast to the earliest clinical trials of boosted atazanavir, these studies included cohorts of patients who were ART experienced at the time atazanavir was administered and who may not have been monitored as closely for VF thus enabling their viruses to evolve for longer period of time under atazanavir selection pressure. Moreover, these studies have included an increasing proportion of sequences from patients with non-subtype B viruses.

Our analysis confirmed that the five major DRMs selected in vitro by atazanavir (V32I, M46I, I50L, I84V, and N88S) were among the most commonly occurring major DRMs. Four additional DRMs also occurred commonly, including M46L, I54V, V82A, and L90M. I50L is a signature atazanavir-associated DRM because it has only been reported in patients receiving atazanavir and it increases susceptibility to PIs other than atazanavir [48,49]. N88S is also considered a signature atazanavir-associated DRM because it is rarely selected by other PIs and it does not significantly reduce susceptibility to other PIs, with the exception of nelfinavir and indinavir [48]. L/M89T may also be a signature atazanavir-associated mutation because it appears to occur more commonly in patients receiving atazanavir than in patients receiving any other PI; for example, it has only been reported in three previously PI-naïve patients receiving lopinavir/r (https://hivdb.stanford.edu/cgi-bin/Mutations.cgi?Gene=PR; accessed on 1 November 2021). In contrast, each of the remaining atazanavir-selected mutations appear to be commonly selected by other PIs, in particular lopinavir/r.

With the exception of L10F and L33F, each of the above 17 most commonly selected major or accessory DRMs was significantly associated with reduced atazanavir susceptibility in a previously published weighted least squares regression analysis of 1644 sequences [48]. Few published phenotypic data are available on sequences containing L89T.

Some limitations of our review should be discussed. First, most of the sequences that we reviewed were obtained from retrospective cohort studies and case series. For these studies, the duration of therapy, accompanying ARVs, frequency of virological monitoring and genotypic resistance testing, and duration of virological failure were generally not available. Therefore, the extent to which these factors were associated with emergent PI-associated DRMs could not be explored. Second, the dataset contained an under-representation of subtypes other than subtype B. Third, we could not be sure that every sequence was obtained from a patient receiving atazanavir as his/her first PI as treatment histories are often incomplete. Nonetheless, we emailed the authors of those studies containing the largest numbers of DRMs and received confirmation that, to the authors’ knowledge, the patients had just received atazanavir. Fourth, at least 32 studies in PubMed that contained sequences from 1000 patients receiving boosted or unboosted atazanavir could not be included in our analysis because the primary sequence data and complete list of protease mutations were not available.

In conclusion, to our knowledge, this is the only comprehensive analysis of atazanavir-selected mutations. Our analysis shows that the spectrum of atazanavir-selected mutations extends beyond those mutations observed in the earliest clinical trials in which patients received either boosted or unboosted atazanavir. The expanded spectrum is likely due to the large number of sequences in our analysis and the likelihood that many of the patients in the studies we reviewed had prolonged VF and ongoing replication while receiving atazanavir. The study also identified one novel nonpolymorphic atazanavir-selected mutation that predominantly occurred in non-subtype B sequences. The relatively low cross-resistance to darunavir/r combined with preliminary data suggests that boosted atazanavir can be an efficacious regimen for second-line therapy. However, comparative clinical trials are required to determine the optimal boosted PI to use for second-line and potentially later-line therapy in LMICs.

## 4. Materials and Methods

### 4.1. Study Selection Criteria

We analyzed publicly available HIV-1 group M protease nucleotide sequences obtained from previously PI-naïve patients receiving boosted or unboosted atazanavir. Sequences were obtained from HIVDB, which is populated with sequences from GenBank annotated with the ART history of the patients from whom the sequences were obtained [13]. The analysis was last updated 31 December 2021. We supplemented the data in HIVDB with previously unpublished sequences performed at SUH and with previously unpublished sequences from two collaborating research groups: the EIDB [14] and the RHIVDB [15]. Additionally, we performed a PubMed search to identify studies describing HIV-1 group M protease sequences that were not present either in HIVDB or GenBank.

Publications reporting eligible protease sequences were reviewed to determine the treatment history of the patient from whom each sequence was obtained to confirm that the patient had received no PI prior to atazanavir and to distinguish those patients receiving unboosted atazanavir from boosted atazanavir. Each sequence was annotated with the year and country of virus isolation, the type of sample (e.g., PBMCs), the sequencing method (Sanger dideoxynucleoside sequencing versus NGS), and the nature of the study population. HIV-1 subtype was determined using the HIVDB subtyping program [50].

We also characterized each study according to whether it included patients in a clinical trial or in a treatment cohort for whom genotypic resistance testing was routinely available for patients with VF as opposed to a case series or case reports for which the indications for genotypic resistance testing were not reported. Studies that performed routine genotypic resistance testing on all patients with VF provide information on how often PI resistance arises in patients receiving atazanavir. In contrast, the remaining studies were considered likely to be enriched for patients with acquired PI resistance.

### 4.2. Mutations

PI-associated DRMs were defined as those with an HIVDB drug resistance program penalty score for ≥1 PI as of December 31, 2021 [51]. The DRMs included 57 mutations at 24 positions: L10F, K20T, L23I, L24I/F/M, D30N, V32I, L33F, K43T, M46I/L/V, I47A/V, G48A/L/M/Q/S/T/V, I50I/L, F53L, I54A/L/M/S/T/V, Q58E, G73A/C/D/S/T/V, T74P, L76V, V82A/C/F/L/M/S/T, N83D, I84A/C/V, N88D/G/S/T, L89V, and L90M. Major mutations were defined as those with a greater effect on the susceptibility to one or more PIs, an increased occurrence in patients with VF on PI-containing regimens, and a low likelihood of occurring without selective drug pressure. Additional PI-associated NP-TSMs that are not classified as DRMs were also examined [52]. The NP-TSMs included 56 mutations at 31 positions: L10R/Y, V11L, K20A, A22V, L33M, E34D/N/Q/R/V, M36A, L38W, K43I/N/P/Q/S, K45I/Q/V, G48E, G51A, F53I/W/Y, K55R/N, I66F/L/V, C67F/L, A71I/L, I72K/L, G73I/N, T74E, P79N, V82G, N83S, I85V, L89P/T, T91C/S, Q92R, C95F/L/V, and T96S.

### 4.3. Analyses

The Fisher’s Exact Test was used to compare the proportion of each mutation in sequences from patients receiving boosted versus unboosted atazanavir, from patients who were previously ART-naïve versus ART-experienced, and from patients according to whether they had subtype B versus non-subtype B sequences. The Wilcoxon Rank Sum Test was used to compare the median number of mutations between two groups. The Holm’s method was used to control for the familywise error rate for multiple hypothesis testing [53].

A binomial regression model was used to examine the relationship between the year of ART initiation and the presence or absence of PI-associated DRMs. To assess the association of covariates with the presence or absence of PI-associated DRMs, a multivariate generalized linear mixed logistic regression analysis was performed using the R package lme4. To account for study heterogeneity, study was included in the model as a random effect.

To identify the patterns of covariation among DRMs and NP-TSMs, we calculated Jaccard similarity coefficients and their standard Z scores for all pair of mutations [54]. To capture conditional dependency among the significantly co-occurring mutation pairs, defined as those pairs that had Jaccard similarity coefficient *p* < 0.01, we constructed a Bayesian network with a hill-climbing search using the R package bnlearn [55] and created a directed edge network graph using the R package visNetwork [56]. To learn the structure of the Bayesian network of core mutations associated with atazanavir, we excluded sequences containing more than four DRMs in this analysis.

For each sequence containing one or more DRMs, we determined the level of predicted resistance to atazanavir and the levels of predicted cross resistance to lopinavir/r and darunavir/r using the HIVDB drug resistance interpretation system.

### 4.4. Accession Numbers

Sequences in this study had been submitted to GenBank (accession numbers ON058287-ON058987).

## Figures and Tables

**Figure 1 pathogens-11-00546-f001:**
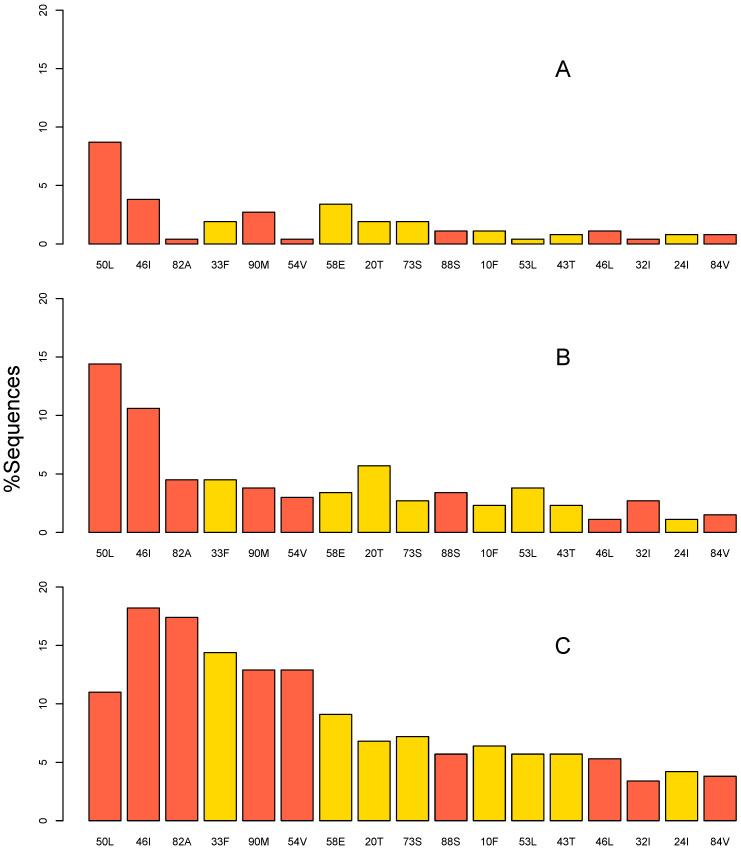
Prevalence of PI-associated drug-resistance mutations (DRMs) in 264 sequences containing 1 or more DRMs from previously PI-naïve patients receiving a boosted or unboosted atazanavir-containing regimen. The distribution of DRMs is plotted separately according to the number of PI-associated DRMs in the sequence: (**A**) 1 DRM, (**B**) 2 to 3 DRMs, and (**C**) ≥4 DRMs. The DRMs shown are those occurring in ≥5% of the sequences, including 9 major DRMs indicated in red and 8 accessory DRMs indicated in yellow.

**Figure 2 pathogens-11-00546-f002:**
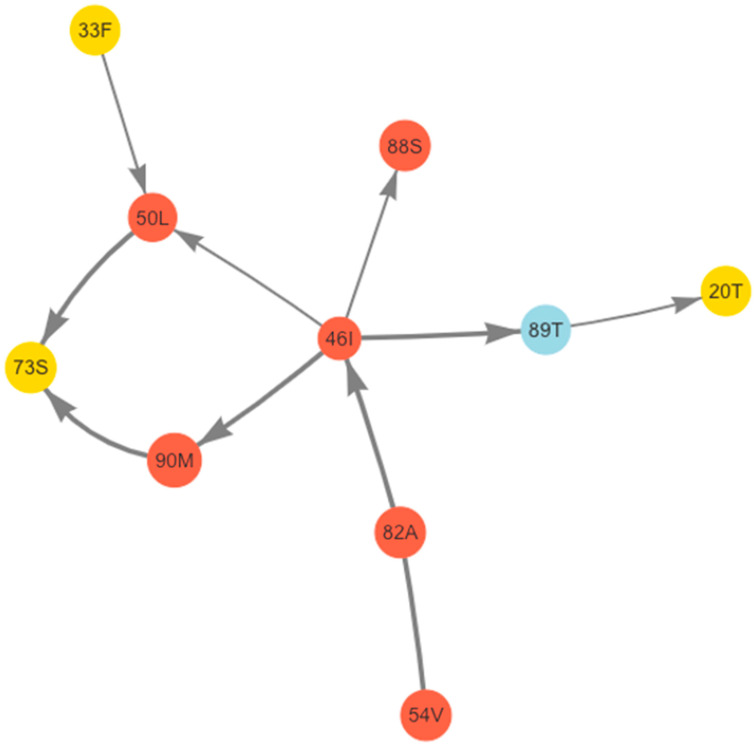
Bayesian network analysis of positively correlated mutation pairs with a hill-climbing search. The Bayesian network analysis yielded 11 mutation pairs, including 6 major DRMs (red), 3 accessory DRMs (yellow), and an additional nonpolymorphic treatment-selected mutation (light blue) with a significant Jaccard correlation coefficient (*p* < 0.01). The thickness of the arrows indicates the strength of the probabilistic relationship of the two mutations. The direction of the probabilistic causation is shown with an arrowhead. For the direction between V82A and I54V for which the probabilistic causation is not greater than the probabilistic causation of the opposite direction by 0.1, the arrowhead is not shown.

**Table 1 pathogens-11-00546-t001:** Studies containing publicly available sequences from previously PI-naïve patients receiving boosted or unboosted atazanavir (ATV).

AuthorYr	Study Type	# Total ATV	# bATV	# ATV	% DRMs ^1^	Median Year	Country	Subtypes (%) ^2^
** *Large clinical trials and cohorts for which genotypic resistance testing was routinely available at virological failure* **								
EuResist Network [14]	Cohort	562	286	276	10.3	2012	Europe	B (57.8), G (16.2), 02_AG (12)
Stanford University Hospital	Cohort	152	142	10	9.2	2010	U.S.	B (96.7)
Mollan12 [16]	ACTG A5202	137	137	0	5.8	2006	U.S.	B (97.1)
Kantor15 [17]	ACTG A5175	117	19	98	14.5	2006	Multi-continents	C (55.6), B (41.9)
Lennox14 [18]	ACTG A5257	69	69	0	2.9	2010	U.S.	B (97.1)
** *Case series and cohorts for which genotypic resistance testing may not have been routinely available at virological failure* **								
Soldi19 [10]	Cohort	149	81	68	30.2	2015	Brazil	B (75.8), F (12.8)
Tarasova21 [15]	Cohort	43	16	27	37.2	2017	Russia	A (90.7)
Kouamou19 [19]	Cohort	40	40	0	12.5	2017	Zimbabwe	C (100)
de Carvalho Lima20 [20]	Cohort	37	28	9	54.1	2010	Brazil	B (81.1), F (16.2)
Acharya14 [21]	Cohort	35	35	0	48.6	2013	India	C (80), A (20)
Ndashimye18 [22]	Cohort	33	33	0	42.4	2016	Uganda	A (57.6), D (24.2), B (15.2)
Gulick04 [23]	ACTG A5095	24	1	23	8.3	2003	U.S.	B (100)
Colonno04 [24]	Case series from clinical trials ^3^	21	0	21	100	2000	Multi-continents	B (71.4), C (28.6)
Chimukangara16 [25]	Cohort	17	17	0	29.4	2015	Zimbabwe	C (100)
Posada Cespedes21 [12]	Cohort	13	7	6	7.7	2015	South Africa	C (100)
Makwaga20 [26]	Cohort	11	11	0	36.4	2020	Kenya	A (63.6), B (18.2), D (18.2)
de Sa Filho08 [27]	Cohort	10	8	2	80	2006	Brazil	B (80), F (20)
Kolomeets14 [28]	Cohort	10	0	10	30	2012	Russia	A (70), 02_AG (30)
Alves19 [29]	Cohort	3	2	1	0	2017	Brazil	C (66.7), B (33.3)
Kim13 [30]	Cohort	3	1	2	33.3	2011	Korea	B (100)
Karkashadze19 [31]	Cohort	2	0	2	100	2015	Republic Of Georgia	A (50), B (50)
Armenia20 [32]	Cohort	1	1	0	0	2012	Italy	B (100)
El-Khatib10 [33]	Cohort	1	1	0	100	2008	South Africa	C (100)
Hoffmann13 [34]	Cohort	1	0	1	0	2010	South Africa	C (100)
Mziray20 [35]	Cohort	1	1	0	0	2018	Tanzania	C (100)
Neogi16 [36]	Cohort	1	0	1	0	2013	South Africa	C (100)
Riddler08 [37]	ACTG A5142	1	1	0	0	2004	U.S.	D (100)
Rosen-Zvi08 [38]	Cohort	1	1	0	0	2006	Germany	B (100)
Svard17 [39]	Cohort	1	1	0	0	2013	Tanzania	A (100)
Vergani08 [40]	Cohort	1	0	1	0	2006	Italy	B (100)

Footnotes: ^1^ DRMs were defined as those with a Stanford HIV drug resistance program penalty score for ≥1 PI. ^2^ Subtypes with ≥10% sequences were listed. ^3^ Colonno04 contained sequences from previously PI-naïve patients with virological failure with resistance on ATV-containing regimens in three clinical trial, AI424-007/041, AI424-008/044, and AI424-034. Additional notes: All studies used the Sanger dideoxynucleoside sequencing method, except for Alves19 in which next-generation sequencing was used; samples from peripheral blood mononuclear cells (PBMCs) were used in Alves19, Makwaga20, and Mziray20, and from both PBMC and plasma in Kim13. In the remaining studies, plasma was used. Abbreviation: b-ATV—boosted atazanavir.

**Table 2 pathogens-11-00546-t002:** Proportion of patients with PI-associated drug resistance mutations (DRMs) and median number of DRMs per patient according to ART history and HIV-1 subtype.

	# Patients, (% of Total; n = 1437)	# Patients with ≥1 DRMs ^1^, (% of Row Total)	Median # DRMs in Patients with ≥1 DRM (IQR)
***Unboosted* vs. *boosted***			
Unboosted	558 (37.3)	117 (21.0)	3.0 (1.0–4.0)
Boosted	939 (62.7)	147 (15.7)	2.0 (1.0–4.0)
***Subtype B* vs. *non-subtype B***			
Subtype B	570 (38.1)	150 (16.2)	3.0 (1.0–4.0)
Non-subtype B	927 (61.9)	114 (20.0)	3.0 (1.0–4.0)
***ART-naïve* vs. *ART-experienced***			
ART-naïve	907 (60.6)	136 (15.0)	3.0 (1.0–4.0)
ART-experienced	590 (39.4)	128 (21.7)	2.0 (1.0–4.0)
** *Year of ART initiation* ** ^2^			
1993–2004	134 (11.9)	24 (17.9)	2.0 (1.0–2.1)
2005–2006	362 (32.1)	44 (12.1)	1.0 (1.0–2.4)
2007–2009	316 (28.0)	26 (8.2)	2.0 (1.0–2.8)
2010–2018	315 (28.0)	29 (9.2)	2.0 (1.0–2.3)

Footnotes: ^1^ DRMs were defined as those with an HIVDB drug resistance program penalty score for ≥1 PI. ^2^ Patients with available year of ART initiation (n = 1127) were grouped into four time periods containing approximately equal numbers of patients.

**Table 3 pathogens-11-00546-t003:** Drug resistance mutations (DRMs) occurring in ≥1 sequences from patients receiving boosted or unboosted atazanavir as their first PI.

DRM ^1^	Classification ^2^	% in the 264 Patients with a PI-Associated DRM	Median # Co-Occurring DRMs (IQR)
I50L	Major	34.1	2 (0.2–3)
M46I	Major	32.6	3 (2–5)
V82A	Major	22.3	4 (3–5)
L90M	Major	19.3	3 (2–4.5)
I54V	Major	16.3	4 (3–5)
N88S	Major	10.2	3 (2–4)
M46L	Major	7.6	3 (2–4)
V32I	Major	6.4	3 (2–5)
I84V	Major	6.1	3 (2–5)
I54L	Major	4.2	3 (3–4.5)
G48V	Major	3.4	3 (2–3)
I47V	Major	2.7	5 (4.5–7)
I50V	Major	2.3	4 (3–5)
L76V	Major	2.3	5.5 (4.2–6)
I47A	Major	1.5	3.5 (2–5)
V82M	Major	1.5	2 (1.7–3)
V82T	Major	1.5	4.5 (3.5–5.5)
D30N	Major	1.1	4 (3.5–7)
G48A	Major	1.1	6 (4.5–6.5)
V82F	Major	1.1	6 (5–6.5)
V82L	Major	1.1	3 (1.5–4.5)
I54A	Major	0.8	3.5 (3.2–3.7)
V82S	Major	0.8	3 (3–3)
G48M	Major	0.4	2 (2–2)
I54M	Major	0.4	5 (5–5)
I54S	Major	0.4	2 (2–2)
I54T	Major	0.4	2 (2–2)
V82C	Major	0.4	4 (4–4)
N88T	Major	0.4	3 (3–3)
L33F	Accessory	20.8	4 (2–5)
Q58E	Accessory	15.9	3 (1–5)
K20T	Accessory	14.4	2 (1–4)
G73S	Accessory	11.7	3 (1–4)
L10F	Accessory	9.8	4 (2–5)
F53L	Accessory	9.8	3.5 (2–5)
K43T	Accessory	8.7	4 (2–5)
L24I	Accessory	6.1	4 (2–4.2)
L23I	Accessory	4.2	3 (1.5–4.5)
T74P	Accessory	3	3 (2–4)
G73T	Accessory	1.5	3.5 (2.7–4.5)
L89V	Accessory	1.5	3.5 (2–5.2)
N83D	Accessory	1.1	3 (3–4)
N88D	Accessory	1.1	3 (2.5–6.5)
G73C	Accessory	0.8	3.5 (2.2–4.7)
L24F	Accessory	0.4	0 (0–0)
M46V	Accessory	0.4	2 (2–2)
G73A	Accessory	0.4	6 (6–6)
G73V	Accessory	0.4	8 (8–8)

^1^ DRMs were defined as those with a Stanford HIVDB drug resistance program penalty score for ≥1 PI. ^2^ See the method for DRM classification.

## Data Availability

Datasets used in the present study are available in Table 1 and Appendix A.

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
