# Peer review of "Spectrum of Atazanavir-Selected Protease Inhibitor-Resistance Mutations"

_pathogens, 2022, doi:10.3390/pathogens11050546_

Round 1
Reviewer 1 Report
The paper by Soo-Yon Rhee et al. is an analysis of protease sequences of genotypic resistance tests performed in patients failing an atazanavir-based antiretroviral regimen, evidencing an overall low incidence of resistance mutations to atazanavir, and making subgroup analysis based on HIV viral subtype and dividing the population into boosted and unboosted atazanavir; also, the expected proportion of resistance to other PIs (lopinavir and darunavir) after developing resistance mutations to atazanavir is included. The sequences were obtained from several clinical trials and cohort studies performed in different Countries and cover a large period of time. Like the authors underline in the Conclusions, this report represents the only comprehensive analysis of the atazanavir-selected mutations, emerging from clinical practice and adds important information to that highlighted during earlier in-vitro studies and clinical trials. Overall, the manuscript’s results are noteworthy and deserve attention; the data collection and the statistical analysis performed are well described.
In my opinion, only few points should be addressed before publication.
Major comments:
- Data from recent clinical trials (see Paton et al., Dolutegravir or Darunavir in Combination with Zidovudine or Tenofovir to Treat HIV; and Aboud et al., Dolutegravir versus ritonavir-boosted lopinavir both with dual nucleoside reverse transcriptase inhibitor therapy in adults with HIV-1 infection in whom first-line therapy has failed (DAWNING): an open-label, non-inferiority, phase 3b trial) underline the results obtained with dolutegravir, lopinavir and darunavir in the context of failure to first-line regimens in low-income Countries. From a clinical perspective, it could be argued that atazanavir does not represent a first option as a second line regimen, especially if failing with this drug could also compromise, at least partly, the efficacy of darunavir. Even if substantial knowledge about this topic is fundamental, a comment should be added to put the clinical meaning of these findings in the context of previously cited studies.
- If data about the atazanavir-containing regimen is available, an effort should be performed to differentiate proportion of mutations between naïve and treatment-experienced patients and, above all, according to the backbone used in association with atazanavir: were there failures with two-drug regimens (for instance, lamivudine plus boosted atazanavir) in the population? and were mutations to atazanavir more frequent with the use of the dual therapy rather than the triple regimens? The addition of these information could explain the reason why mutations were less frequent in the boosted atazanavir group (since it reflects a more accurately selected population), and could invite clinicians to the use of simplified treatment regimens even in low and middle-income Countries.
Minor comments:
- in the abstract, the acronym HIVDB (line 27) should be explained
- In the abstract, it should be specified that the sequences analyzed come from patients with virological failure during an atazanavir-containing regimen.
Author Response
Major comments:
- Data from recent clinical trials (see Paton et al., Dolutegravir or Darunavir in Combination with Zidovudine or Tenofovir to Treat HIV; and Aboud et al., Dolutegravir versus ritonavir-boosted lopinavir both with dual nucleoside reverse transcriptase inhibitor therapy in adults with HIV-1 infection in whom first-line therapy has failed (DAWNING): an open-label, non-inferiority, phase 3b trial) underline the results obtained with dolutegravir, lopinavir and darunavir in the context of failure to first-line regimens in low-income Countries. From a clinical perspective, it could be argued that atazanavir does not represent a first option as a second line regimen, especially if failing with this drug could also compromise, at least partly, the efficacy of darunavir. Even if substantial knowledge about this topic is fundamental, a comment should be added to put the clinical meaning of these findings in the context of previously cited studies.
- If data about the atazanavir-containing regimen is available, an effort should be performed to differentiate proportion of mutations between naïve and treatment-experienced patients and, above all, according to the backbone used in association with atazanavir: were there failures with two-drug regimens (for instance, lamivudine plus boosted atazanavir) in the population? and were mutations to atazanavir more frequent with the use of the dual therapy rather than the triple regimens? The addition of these information could explain the reason why mutations were less frequent in the boosted atazanavir group (since it reflects a more accurately selected population), and could invite clinicians to the use of simplified treatment regimens even in low and middle-income Countries.
We agree with the statement that “atazanavir does not represent a first option as a second line regimen”. And because we do not want to convey the impression that we are promoting atazanavir, we have changed the abstract as follows: “Ritonavir-boosted atazanavir is an important second line therapy option in low- and middle-income countries” to “Ritonavir-boosted atazanavir is an option for second line therapy in low- and middle-income countries (LMICs)”.
The reviewer makes a logical request which has prompted us to include two types of additional data in the Results: (i) the proportion of patients who were ART-naïve before receiving atazanavir; and (ii) the accompanying ARVs administered with atazanavir. As a result of this suggestion and a request by Reviewer 3 (see below), we have also provided information on whether the year of ART initiation also influenced the risk that a patient’s sequence would contain a PI-associated DRMs. We have provided this information in a new Table (Table 2) and in a new section of the Results (2.5 ART experience). In addition, we have added a multivariate regression analysis in an attempt to identify potential variables associated with the emergence of a PI-associated DRM.
The new section in the Results as follows:
“2.5. ART experience
Among the 1,497 patients receiving atazanavir, 907 (60.6%) were previously ART-naïve and 590 (39.4%) were ART-experienced (Table 2). The proportion of sequences containing one or more PI-associated DRMs was 21.7% in previously ART-experienced patients and 15.0% in previously ART-naïve patients (p=0.001). Among those with one or more PI-associated DRMs, the number of DRMs was not significantly different in previously ART-experienced patients (median 2 DRMs; IQR: 1-4 DRMs) compared with previously ART-naïve patients (median 3 DRMs; IQR: 1-4 DRMs; p=0.3).
Among the 907 previously ART-naïve patients, atazanavir was administered with two nucleoside RT inhibitors (NRTIs) in 840 (92.6%) patients. Among the remaining 67 patients, the co-administered antiretroviral drugs (ARVs) were not provided for 44 (4.9%) while 23 (2.5%) received a variety of other ARVs.
Among the 590 previously ART-experienced patients, atazanavir was administered with two NRTIs in 345 (58.5%) patients. Among the remaining 245 patients, the co-administered ARVs were not provided for 163 (27.6%) while 82 (13.9%) received a variety of other ARVs. Only four patients received atazanavir plus one additional ARV.
The year of ART-initiation was available for 1127 (75.3%) of all patients. The patients could be pooled into four time periods containing approximately equal numbers spanning the years between 1993 and 2018 (Table 2). The proportion of patients with one or more PI-associated DRMs decreased over time (binomial coefficient = -0.26; 95% CI: -0.45 to -0.07; p=0.007) but the number of DRMs in patients with one or more DRMs did not change.
Using just those patients for whom the year of ART initiation was available, a multivariate linear regression analysis was performed to assess the association between four factors and the development of a PI-associated DRM. The four factors included the year of ART initiation, subtype (B vs. non-subtype B), the use boosted vs. unboosted atazanavir, and previous ART (naïve vs. experienced). The analysis found that a later year of ART initiation (OR: 0.62; 95%CI: 0.49-0.79; p=0.0001) and the administration of boosted atazanavir (OR: 0.57; 95%CI: 0.35-0.93; p=0.02) were associated with a decreased risk of developing a PI-associated DRM.”
Minor comments:
- in the abstract, the acronym HIVDB (line 27) should be explained. This was done.
- In the abstract, it should be specified that the sequences analyzed come from patients with virological failure during an atazanavir-containing regimen. This was done.
Reviewer 2 Report
Rhea S-Y et al looked at the prevalence of DRMs due to ritonavir-boosted or non-boosted atazanavir in PWH using publicly available data. The introduction, study design, methods and results are sound.
I have only one minor comment to make regarding the abstract. The authors summarised the implications of their findings for ART choice in LMICs in the last paragraph of the "Discussion" section. The authors should add a line or two about this in the abstract.
Author Response
I have only one minor comment to make regarding the abstract. The authors summarised the implications of their findings for ART choice in LMICs in the last paragraph of the "Discussion" section. The authors should add a line or two about this in the abstract. We have added the following sentence to the end of the Abstract: “Clinical studies, however, are required to determine the optimal boosted PI to use for second line and potentially later line therapy in LMICs.”
Reviewer 3 Report
In this carefully conducted resistance assessment for atazanavir the authors utilise an impressive body of relevant sequencing information to confirm and complete the current resistance landscape for this PI.
This is of high interest especially for the large settings of LMI countries, especially in Africa, where atazanavir continues to be one of the critical therapy options for HIV.
The manuscript is clearly written and well-structured; key findings but also shortcomings are appropriately discussed.
--> As mentioned by the authors, the unavailability of accompanying data about prior therapy or treatment duration but also about VL or failure information does not allow any further specification of possible links between certain resistance mutations or patterns and a corresponding treatment situation.
However, it could be interesting to add in the failure analysis the function of "therapy years" (eg. early versus recent, maybe by 5-year brackets)). The reasoning is that in earlier years a prior exposure to other PIs, including IDV or SQV, NFV etc, would have been more likely, and which could have contributed to some peculiar resistance patterns. On the other side, this would be less the case in later years, where LPV and ATV (partly DRV) would have become the sole options. - Will there be a difference between these two somewhat different time periods?
This information could be of importance for the treating physician in resource-limited settings as it would advise for acompanying adherence counselling etc.
minor points:
line 158: would the thickness not rather refer to the "arrows"? it is unclear which kind of "edge" could be meant.
line 211: no comma behind "contrast"
Author Response
As mentioned by the authors, the unavailability of accompanying data about prior therapy or treatment duration but also about VL or failure information does not allow any further specification of possible links between certain resistance mutations or patterns and a corresponding treatment situation. We have partially addressed this limitation by adding the section 2.5 entitled: “ART experience” as outlined above in response to Reviewer 1.
However, it could be interesting to add in the failure analysis the function of "therapy years" (eg. early versus recent, maybe by 5-year brackets)). The reasoning is that in earlier years a prior exposure to other PIs, including IDV or SQV, NFV etc, would have been more likely, and which could have contributed to some peculiar resistance patterns. On the other side, this would be less the case in later years, where LPV and ATV (partly DRV) would have become the sole options. - Will there be a difference between these two somewhat different time periods? This information could be of importance for the treating physician in resource-limited settings as it would advise for acompanying adherence counselling etc. This is a very helpful suggestion. In response to this reviewer and reviewer #1, we have added a new section (2.5. ART experience) in the Results that describing the relationship between ART-experience, accompanying antiretrovirals, and the year therapy was begun to the risk of containing one or more PI-associated DRMs and to the median number of DRMs.
Minor points:
- line 158: would the thickness not rather refer to the "arrows"? it is unclear which kind of "edge" could be meant. We have changed it to “arrows”.
- line 211: no comma behind "contrast". The comma has been removed from the sentence.
Round 2
Reviewer 1 Report
The work has been improved and I believe that is now fully suitable for publication.
I only invite the authors to modify line 229: "a multivariate linear regression analysis was performed..." with "a multivariable logistic regression analysis was performed". I think the outcome is a dichotomous variable.